# Emerging Precision Medicine Approaches for Lung Neuroendocrine Tumors

**DOI:** 10.3390/cancers15235575

**Published:** 2023-11-25

**Authors:** Claire K. Mulvey

**Affiliations:** Division of Hematology/Oncology, Department of Medicine, University of California, San Francisco, CA 94158, USA; claire.mulvey@ucsf.edu

**Keywords:** lung neuroendocrine tumors, carcinoid tumors, precision medicine, theragnostics

## Abstract

**Simple Summary:**

Precision medicine is a new approach to the diagnosis and treatment of cancer that considers individual patient differences and avoids a one-size-fits-all approach to treatment. Lung neuroendocrine tumors are increasingly common cancers with a wide spectrum of clinical behaviors. Lung neuroendocrine tumors have lagged behind other cancers in the incorporation of precision medicine approaches to treatment. However, this is beginning to change as our understanding of the molecular underpinnings of neuroendocrine tumor carcinogenesis expands and our clinical trial repertoire increases. In particular, advances in radiotheragnostics hold promise for yielding new personalized approaches to treatment. In this review, we summarize established and emerging precision medicine approaches to the treatment of lung neuroendocrine tumors.

**Abstract:**

Well-differentiated lung neuroendocrine tumors (LNETs) are heterogeneous cancers that are increasing in incidence. Treatment options for LNETs have expanded in recent years, and our knowledge of the molecular subtypes has also advanced. Multidisciplinary teams have an established role in personalizing the best treatment for individual patients. Other precision medicine approaches for the treatment of LNETs have lagged behind those for non-small-cell lung cancer, with only rare actionable molecular alterations identified and few established predictive factors to guide therapy selection. However, as summarized in this review, there is increasing potential for personalized treatment of patients with LNETs. In particular, advances in radiotheragnostics may allow us to tailor the treatment of individual patients with NETs in the coming years. These advances may soon deliver the promise of more effective, less toxic treatments and better outcomes for patients with these increasingly common cancers.

## 1. Introduction

Lung neuroendocrine neoplasms (LNENs) are an increasingly common, heterogeneous group of cancers that arise from enterochromaffin cells. NENs are characterized by their site of origin, histologic grade, and ability to secrete peptide hormones. The histologic classification of LNENs incorporates morphology (well-differentiated neuroendocrine tumors (NETs) versus poorly differentiated neuroendocrine carcinomas (NECs)), mitotic count, and the presence or absence of necrosis [1]. Well-differentiated LNETs are further subdivided into typical carcinoid (grade 1) and atypical carcinoid (grade 2) tumors. Both typical and atypical carcinoids have a different morphology from the poorly differentiated neuroendocrine carcinomas of the lung: small-cell lung carcinoma (SCLC) and large-cell neuroendocrine carcinomas (LCNECs). However, these malignancies exist on a spectrum, and some cases exist in a gray zone with well-differentiated morphologies but elevated mitotic counts.

LNETs account for approximately 2% of all lung cancers in adults [2]. The lung is the second most common NET primary site after the gastrointestinal tract [3]. LNETs typically develop between the fourth and sixth decade of life, and most series suggest a higher incidence in women compared with men and in White compared with Black or Asian populations [3,4,5,6,7,8]. The incidence of LNETs appears to be increasing over time. The reasons for this rising incidence are not entirely clear. Improved cross-sectional imaging and bronchoscopy techniques may be contributing, along with better recognition of the histology by pathologists and increased awareness among clinicians.

While the more common non-small-cell lung cancers (NSCLCs) have seen an explosion of precision medicine approaches to treatment over the last decade, LNETs have lagged behind. Recently, however, progress has been made in identifying molecular characteristics and biological subsets of LNETs that may pave the way for more personalized treatment approaches in the future. In addition, randomized controlled trials have become more common in LNETs, expanding our knowledge about how to treat patients and increasing the number of therapeutic options. Advances in molecular imaging and radiotheragnostics, in particular, have revealed the power and potential of personalized treatments for NETs. In this review, we discuss established and emerging precision medicine approaches for the treatment of well-differentiated LNETs, distinct from the high-grade NECs of the lung.

## 2. Multidisciplinary Evaluation and Clinical Precision Medicine

Well-differentiated LNETs have a wide spectrum of clinical behaviors. Some patients with advanced LNETs (usually those with typical carcinoids) have very indolent disease, while other patients (usually those with atypical carcinoids) can have a more aggressive course. There is both the possibility of interpatient and intrapatient tumor heterogeneity, with variable somatostatin receptor expression, molecular changes, and Ki-67 proliferation indices. There is also increasing appreciation for the challenges of grade progression, where some NETs, particularly those of pancreatic origin, can evolve over time to a more aggressive form, necessitating treatment changes [9].

Standard systemic therapy options for LNETs include somatostatin analogs (SSAs); targeted therapies like everolimus; and chemotherapies, including capecitabine/temozolomide or platinum/etoposide. There are limited randomized data on how to sequence these available therapies. Due to their rarity, patients with LNETs have been excluded from most clinical trials of NET therapies, and treatment principles are largely extrapolated from gastroenteropancreatic (GEP) NETs. Treatment decisions for patients with LNETs are complex and must take into account several different aspects of a patient’s presentation, including histologic features, tumor growth rate, degree of symptoms, and hormone hypersecretion, along with any concerning molecular features (e.g., loss of *RB1* and/or *TP53*) or radiographic features (e.g., relative uptake in FDG versus somatostatin receptor scintigraphy). Multidisciplinary discussions are essential given the wide range of clinical behaviors. The multidisciplinary approach to assessing patients with LNETs is itself a form of clinical precision medicine, whereby the team synthesizes multiple aspects of a given patient’s presentation to determine the best-individualized treatment strategy. International guidelines for NETs emphasize the importance of collaboration among medical subspecialties as a cornerstone to deliver optimal, personalized care [10,11].

## 3. Molecular Characterization of LNETs

Next-generation sequencing (NGS) is frequently used in NSCLCs to identify actionable genomic alterations and select patients to treat with targeted therapies, but such approaches have not been widely used for NETs. However, interest has grown in identifying reliable biomarkers to help differentiate subgroups of NETs and enable more personalized treatment approaches. Over the last decade, the molecular and clinical classification of NETs has evolved. Mutational analysis of LNETs via high-throughput sequencing approaches has provided insight into their molecular characteristics. Whole-genome sequencing revealed that well-differentiated NETs have a very low mutation rate compared with many other malignancies, and they are characterized by mutations in genes involved in histone modification/chromatin remodeling (e.g., *MEN1* and *ARID1A*) and telomere maintenance [12,13]. There is also increasing appreciation for the role of epigenetic modifications and dysregulation in driving NET evolution [14]. Hypermethylation of several tumor suppressor genes, such as *RASSF1* and *p15INK4b*, has been described in LNETs [15,16], along with loss of histone 4 acetylation at lysine 16 and trimethylation at lysine 20 [17]. Down the line, these mechanisms might be targetable via novel treatments. Although the rate of actionable genomic alterations is generally low, several alterations have been described rarely in LNETs, especially atypical carcinoids, including ALK fusions [18], EGFR mutations [19], and RET fusions [20]. Comprehensive molecular profiling with NGS is reasonable in patients with metastatic LNETs to identify rare but actionable alterations that could open up additional approaches for personalized treatment.

While the WHO pathologic classification system has been useful for standardizing the nomenclature of NETs and helping predict clinical behavior, it does not fully reflect the biological heterogeneity of NETs. There is increasing recognition of the existence of lung tumors with well-differentiated carcinoid morphologies but elevated mitotic counts of >10 per 2 mm^2^ and/or Ki-67 of >20% [21]. These highly proliferative carcinoids may not be easily distinguished from NECs based on morphology or mitotic count; for example, cases may have heterogeneous patterns of mitoses with hot spots of high proliferation interspersed with low-proliferative areas [2]. NET classification from bronchoscopic biopsy samples can be especially challenging, as crush artifacts can alter the appearance of carcinoids to mimic SCLCs. Comprehensive next-generation sequencing may be informative in these cases to help sort out the underlying diagnosis and clarify expected clinical behavior. Molecular data suggest that these highly proliferative carcinoid tumors tend to have mutations (e.g., *MEN1*) more characteristic of typical carcinoids rather than *RB1* or *TP53* alterations that are typical of NECs [22]. Conversely, the presence of co-alterations in *RB1* and *TP53* would suggest an underlying diagnosis of a neuroendocrine carcinoma [23].

Rather than constituting one single pathologic entity, more advanced multi-omics testing suggests that there may actually be several distinct biologic subtypes of lung carcinoids based on variations in gene expression, methylation, and mutational profiles [22,24]. For example, one analysis was able to separate atypical carcinoids and LCNECs of the lung into three distinct and clinically relevant subgroups based on transcriptomic and genomic profiling [25]. These data provide insights into the underlying biology and hold promise for further stratifying LNETs into different prognostic groups with disparate survival outcomes. The integration of genomic and transcriptomic data is not yet incorporated into routine clinical practice, but down the line, molecular studies may serve as the basis for the selection of treatment approaches.

## 4. Radiotheragnostics

Neuroendocrine tumors are unique among cancers in that the most widely implemented predictive test for treatment response is not genomic analysis of tumor tissue, but rather functional imaging. The advent of radiotheragnostics like peptide receptor radionuclide therapy (PRRT), which combines diagnostic imaging with targeted therapies, has transformed the treatment of NETs.

Neuroendocrine tumors have the unique property of somatostatin receptor (SSTR) expression, a feature that was exploited for the development of synthetic SSAs. The possibility of labeling SSAs with radionuclides emerged in the early 1990s. Although other agents were initially used, the therapeutic breakthrough came with the development of beta-emitters coupled to DOTA-chelated compounds, most notably lutetium-177 (^177^Lu)-DOTATATE. ^177^Lu-DOTATATE is a radiopharmaceutical that binds to SSTRs and allows for the targeted delivery of radiation to neuroendocrine cancer cells systemically with high precision, resulting in DNA single-strand breaks and activating programmed cell death. ^177^Lu-DOTATATE was approved for the treatment of GEP-NETs based on the results of the NETTER-1 trial in patients with progressive SSTR-positive mid-gut NETs [26].

Currently, patient selection for PRRTs is based primarily on SSTR-positron emission tomography (PET), which is used to identify patients with high SSTR expression who are most likely to benefit from treatment. ^177^Lu-DOTATATE has been studied primarily in GEP-NETs and is not approved specifically for LNETs, although it is included as a potential therapeutic option in national comprehensive cancer center guidelines for SSTR positive LNETs. Unlike GEP-NETs, which reliably express SSTRs, the SSTR expression in LNETs is more variable [27]. In one study of 56 patients with LNETs who underwent contemporaneous SSTR- and FDG-PET imaging, almost half were potentially not suitable for PRRT [28]. However, although experience with ^177^Lu-DOTATATE in LNETs is more limited, several retrospective studies have reported potential benefits in patients with LNETs, with varying response rates [29,30]. Ongoing prospective clinical trials will better characterize the role of ^177^Lu-DOTATATE in LNETs. There is also interest in using SSTR expression via immunohistochemistry from biopsy tissue as another potential molecular biomarker for PRRT responses for LNETs [31]. Discordance between SSTR expression in imaging and histology has been reported, so this might be a way to identify patients who would still benefit from PRRT even with equivocal SSTR functional imaging [32]. However, SSTR expression via immunohistochemistry may not necessarily identify a functional receptor. For now, this remains experimental and of unclear benefit beyond imaging selection.

Looking to the future, several efforts are underway to better select patients and improve the administration of PRRT. Personalized radiation dosimetry is one area of active research. At present, ^177^Lu-DOTATATE is typically administered with fixed doses of injected radioactivity over a set number of treatment cycles regardless of the extent and distribution of a given patient’s metastases. Individual patients absorb differing doses into both tumors and healthy organs, reflecting the difficulty in predicting how the radiopharmaceutical is distributed, metabolized, and excreted across individuals. Patient-specific dosimetry is the process of quantifying the absorbed dose in different regions of the body from the post-treatment SPECT/CT images obtained after each dose. Dosimetry holds promise for potentially improving PRRT outcomes by ensuring both that the absorbed tumor dose is sufficient and also ensuring that the absorbed dose by healthy organs (e.g., kidneys and bone marrow) does not exceed safe limits over time [33]. Down the line, dosimetry information may contribute to the planning of serial treatments and allow us to better tailor PRRT treatment schedules for individual patients to avoid toxicity, optimize benefits, and improve outcomes.

Separately, clinical tools are being validated to help inform PRRT treatment decisions. The clinical score (CS) is one prognostic score that has potential utility in patients with well-differentiated NETs being considered for ^177^Lu-DOTATATE treatment [34]. This score consists of five categories: available treatments for the tumor type outside of ^177^Lu-DOTATATE, prior systemic treatments, symptoms, tumor burden in critical organs, and the presence of peritoneal carcinomatosis. In an analysis of 248 patients with well-differentiated NETs of a variety of primary sites, an increasing CS was associated with a worse progression-free survival [35]. There was also a suggestion that patients with a CS of ≤ 4 were more likely to benefit from PRRT.

Another area of research is using molecular tests, particularly those that incorporate growth factor signaling and tumor metabolism, to better predict responders versus non-responders to PRRT [36]. Efforts are underway to use blood-based transcriptomic assays or combinations of gene expression and single-nucleotide polymorphism assessments, analyzed with machine learning algorithms, to enhance PRRT efficacy and safety. The PRRT predictive quotient (PPQ) is one such strategy, an algorithm that integrates gene expression data and Ki-67 values to predict responders and non-responders to PRRT [37]. Another strategy is NETest, a multigene assay based on specific NET circulating gene transcripts. The NETest score may reflect real-time tumor activity and serve as a surrogate for the radiographic response to PRRT, with the advantage of being analyzable significantly earlier than current imaging protocols [38]. These tests may help optimize patient selection for PRRT and monitor responses to treatment, potentially allowing a change in treatment strategy for patients deemed unlikely to respond. For now, however, these biomarker tests are not widely available and have yet to be incorporated into routine clinical practice.

The time interval between consecutive PRRT treatment cycles is also under study. It may be that the treatment interval should differ depending on a given patient’s proliferation index/Ki-67 to better target the tumor cells in the active phase of the cell cycle that may be most susceptible to PRRT. Lastly, another investigational approach is intra-arterial administration of PRRT via injection of ^177^Lu-DOTATATE directly into the hepatic artery to achieve a particularly high concentration within liver metastases for patients with liver-dominant disease to improve treatment outcomes.

## 5. Personalized Therapeutic Approaches for LNETs

Next, we highlight recent and potential therapeutic advances for the treatment of LNETs, focusing on emerging precision medicine approaches and active clinical trials enrolling patients with LNETs (summarized in Table 1).

### 5.1. Tyrosine Kinase Inhibitors

Several protein and tyrosine kinase pathways appear to be dysregulated in NET carcinogenesis, which has provided a basis for targeted treatment approaches. The mammalian target of rapamycin (mTOR) signaling pathway plays a key role in cell growth, metabolism, and apoptosis in cancer cells, and dysregulated mTOR signaling contributes to NET development [39,40]. The mTOR inhibitor, everolimus, was approved for the treatment of well-differentiated LNETs based on the results of the RADIANT-4 trial [41]. In addition, NETs are highly vascularized and associated with the overexpression of vascular endothelial growth factor (VEGF) and its receptors [42]. Anti-VEGF receptor tyrosine kinase inhibitors (TKIs) have been evaluated in NETs with some success. Sunitinib, a multitargeted TKI, is approved for treating pancreatic NETs [43]. In recent years, there has been renewed interest in other multitarget TKIs for the treatment of NETs. One agent under study is cabozantinib, a potent inhibitor of VEGF receptor 2, along with mesenchymal-epithelial transition factor (MET), AXL, and RET. Activation of the MET signaling pathway may play a role in NET proliferation, and preclinical models suggest that combined anti-VEGF and anti-MET inhibition may enhance the inhibition of angiogenesis, tumor invasion, and metastasis [44]. The CABINET phase III study was designed to evaluate the activity of cabozantinib in patients with advanced pancreatic and extra-pancreatic NETs after progression on everolimus [45]. Positive interim data from the CABINET study were just announced, prompting the unblinding and early discontinuation of the study by the data and safety monitoring board. We await the publication of the results, but cabozantinib may soon represent a new therapeutic option for patients with progressive NETs, including those that start in the lungs. Combination studies of TKIs are ongoing, including combinations of cabozantinib with either temozolomide chemotherapy (CABOTEM) or nivolumab (Table 1). In addition, several other multitargeted TKIs are being evaluated in clinical trials, including lenvatinib (a potent inhibitor of VEGF 1-3, fibroblast growth factor receptor (FGFR1), platelet-derived growth factor receptor alpha (PDGFR*α*), stem cell factor receptor (KIT), and RET) [46] and surufatinib (targets VEGFR 1-3, FGFR1, and colony-stimulating factor-1 receptor) [47], either alone or in combinations with immunotherapy.

In the future, correlative studies to identify predictive biomarkers for the outcomes of treatment with TKIs will be key to identifying the patients most likely to benefit. In addition, much like for NSCLCs, there is interest in using circulating tumor DNA changes over time to monitor responses to various treatments (including TKIs) to identify patients likely to respond. This approach is under study in clinical trials.

### 5.2. Chemotherapy

LNETs generally have low proliferation indices, so cytotoxic chemotherapy often has limited effectiveness. LNETs may be less sensitive to chemotherapy than pancreatic NETs, but potentially more sensitive than small bowel NETs. Chemotherapy can be considered for LNET patients with rapidly progressive and/or high-volume disease. SCLC regimens like platinum/etoposide are often utilized, particularly for atypical carcinoids with more aggressive pathologic features such as elevated Ki-67 and/or mitotic counts. Several small retrospective studies on metastatic LNETs suggest response rates to platinum/etoposide between 20 and 25% [48,49,50]. There are also limited data to support temozolomide-based approaches for those LNETs with slightly more indolent growth. Temozolomide monotherapy has been shown to have some clinical activity in small numbers of patients with LNETs [51,52]. In addition, the combination of oral temozolomide and capecitabine also had activity in a retrospective case series of LNETs [53,54]. An ongoing clinical trial is evaluating the combination of temozolomide plus cabozantinib.

There is also interest in identifying biomarkers of susceptibility to temozolomide and other alkylator therapy. Temozolomide's mechanism of action is via the alkylation/methylation of DNA, causing base pair mismatch and subsequent tumor cell death. The enzyme O6-methylguanine-DNA methyltransferase (MGMT), which repairs DNA damage mediated by temozolomide, has emerged as a potential biomarker of alkylator therapy responses. MGMT can remove the alkyl group, allowing tumor cells to resist the effects of treatment [55]. The expression of MGMT is dependent on its promoter methylation status, which has been used as a predictive biomarker for sensitivity to alkylating agents in glioblastoma, but not other cancer types [56]. In a prospective clinical trial evaluating the combination of capecitabine and temozolomide in patients with advanced pancreatic NETs, MGMT deficiency was associated with improvement in progression-free survival [57]. While MGMT testing is not widely available and not yet validated for use in standard clinical practice for NETs, this may become an approach down the line to predict temozolomide chemotherapy efficacy. Further study is also needed to clarify the best method for evaluating MGMT activity, as both immunohistochemistry and PCR for promoter methylation have been proposed.

### 5.3. Immunotherapy

The most important treatment breakthrough in extensive-stage SCLC in the last several decades was the addition of immunotherapy to platinum/etoposide chemotherapy as a new standard first-line treatment based on the results of the Impower133 and Caspian trials [58,59]. In contrast, immunotherapy has more limited utility in well-differentiated NETs, and novel approaches to target the immune system are needed. The interest in immunotherapy approaches to treating NETs goes back several decades, ever since interferon-alpha was shown in older studies to have some anti-tumor efficacy and also to improve symptoms of carcinoid syndrome [60]. Interferon has been included as a treatment option in consensus treatment guidelines for advanced NETs over the years [61]. Practically, however, its use was limited by significant toxicity. More recently, the results of studies investigating monotherapy with anti-PD(L)1 antibodies in NETs have been largely disappointing, with low overall response rates. Limited data do suggest there may be some activity of dual-checkpoint inhibition in LNETs, especially atypical carcinoids. CA209-538 was a prospective trial in Australia of the combination of ipilimumab and nivolumab in advanced rare cancers that included patients with advanced NETs [62]. In this study, three of the nine patients with atypical carcinoid LNETs had an objective response; however, levels of potentially predictive biomarkers of immunotherapy response like tumor mutational burden were not reported. Other experimental personalized immunologic approaches include anti-tumor vaccines. An ongoing phase I trial is studying the effects of a survivin long peptide vaccine (SurVaxM) in patients with metastatic NETs with confirmed survivin expression. There is also interest in expanding the role of adoptive cell therapies for NETs. An anti-SSTR chimeric antigen receptor (CAR) T cell directed against SSTRs was evaluated in preclinical models, although there are not yet published data in humans [63]. There are also ongoing efforts to generate autologous tumor-infiltrating lymphocytes (TILs) in a variety of solid tumors that include neuroendocrine tumor patient cohorts. These approaches are interesting but not yet ready for prime time. More research is needed to identify biomarkers to help predict responses to immunotherapy and clinical outcomes in LNETs.

### 5.4. Novel PRRT Compounds

Following the success of ^177^Lu-DOTATATE for NETs, several novel PRRT treatment approaches are under study (Figure 1). Not all patients treated with ^177^Lu-DOTATATE respond, and even for patients whose cancer responds, progression is typically observed in the years following treatment. There is interest in new agents and approaches to overcome these limitations. One area of particularly active investigation is alpha-emitting PRRT [64,65,66]. Compared with beta particles, alpha particles have higher linear energy transfer. Radionuclides that emit alpha particles may cause more damage to targeted cancer cells through the creation of double-stranded rather than single-stranded DNA breaks. In addition, because of their shorter path length, alpha-emitting PRRT may also result in less damage to surrounding healthy tissues. Among the medically relevant alpha-emitters, the radionuclide 225-actinium is considered the most promising, and ^225^Ac-PRRT is actively under study in clinical trials. This may represent an improved therapeutic option for patients with NETs, including those resistant to beta-emitting PRRT. Initial efforts have focused on GEP-NETs and SCLC, so utility in well-differentiated LNETs will have to be investigated down the line.

There has also been renewed interest in SSTR antagonists for PRRT. All SSTR-targeting radiopharmaceuticals in current clinical use (including ^177^Lu-DOTATATE) are SSTR agonists, which bind to receptors and are subsequently internalized and accumulate within cancer cells. Initially, it was thought that SSTR agonists would be more suitable therapeutic agents compared with antagonists because of this internalization, which brings the radioactivity close to the cell’s DNA. SSTR antagonists stay bound on the cell surface and are not internalized. However, recent data suggest that despite the fact that SSTR antagonists internalize poorly into tumor cells, they may actually have higher tumor uptake, possibly because a larger number of receptor binding sites may be recognized by antagonists compared with agonists [67,68]. By binding to more sites on NET cells, SSTR antagonists may further improve treatment responses, particularly when the density of SSTRs is low. SSTR antagonists such as ^177^Lu-satoreotide tetraxetan are now being investigated as a promising new approach both for imaging and therapeutic purposes in NETs. This may open up the possibility of treating tumors with relatively lower SSTR expression, including patients with LNETs with insufficient uptake for standard PRRT.

Another approach under consideration to try to improve PRRT efficacy is through conjugating DOTATATE to Evans blue analogs, which reversibly bind to circulating serum albumin [69]. This may extend the effective half-life of DOTATATE in the blood, improving the pharmacokinetics and increasing accumulation and retention in tumors, although a potential limitation is a corresponding increase in toxicity. Finally, combination approaches that combine PRRT with immune checkpoint inhibitors, DNA damage repair inhibition (e.g., PARP inhibitors), or inhibitors of cellular growth, among others, are under study, largely in GEP-NETs. It remains to be seen whether these strategies hold promise for LNETs as well.

## 6. Conclusions and Future Directions

Although treatment of patients with LNETs remains a challenge, the number of available therapies has increased. Over the last several years, progress has been made toward developing personalized treatment approaches for patients with LNETs. Still, there remains an unmet need for biomarkers to help predict the effectiveness of available therapies. Novel personalized therapeutic approaches hold promise to improve outcomes and will hopefully allow us to better treat patients with these increasingly common malignancies in the near future.

## Figures and Tables

**Figure 1 cancers-15-05575-f001:**
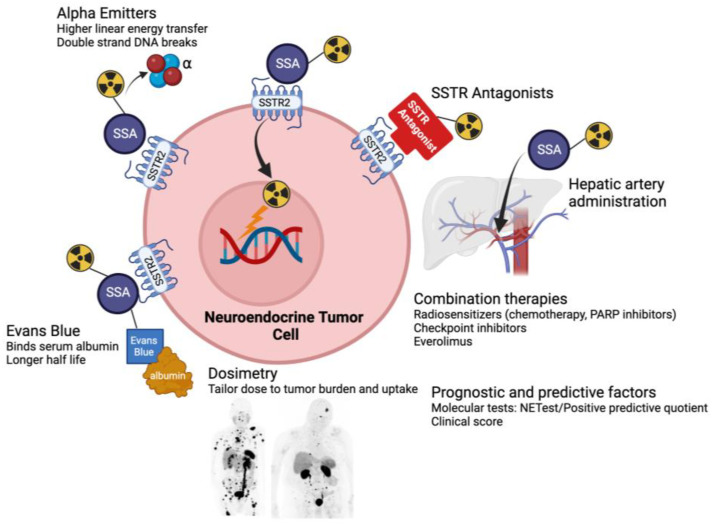
New frontiers in peptide receptor radionuclide therapy. SSA = somatostatin analog; SSTR = somatostatin receptor.

**Table 1 cancers-15-05575-t001:** Select ongoing therapeutic studies on LNETs.

Trial Name	Sponsor City, Country/Region	Phase	Treatment Regimen
TKI Studies and Combination Studies			
A Trial Evaluating the Activity and Safety of Combination Between Cabozantinib and Temozolomide in Lung and GEP-NENS Progressive After Everolimus, Sunitinib or PRRT (CABOTEM)	National Cancer Institute, Naples Naples, Italy	Phase II	Cabozantinib + temozolomide
Testing Lutetium Lu 177 Dotatate in Patients With Somatostatin Receptor Positive Advanced Bronchial Neuroendocrine Tumors	National Cancer Institute Bethesda, USA	Phase II	177Lu-DOTATATE vs. everolimus
Testing Cabozantinib in Patients With Advanced Pancreatic Neuroendocrine and Carcinoid Tumors	National Cancer Institute Bethesda, USA	Phase III	Cabozantinib vs. placebo
Cabozantinib and Nivolumab for Carcinoid Tumors	Dana-Farber Cancer Institute Boston, USA	Phase II	Cabozantinib + nivolumab
Lenvatinib and Everolimus in Treating Patients With Advanced, Unresectable Carcinoid Tumors	MD Anderson Cancer Center Houston, USA	Phase II	Lenvatinib + everolimus
Phase II Study of Pembrolizumab and Lenvatinib in Advanced Well-differentiated Neuroendocrine Tumors	H. Lee Moffitt Cancer Center and Research Institute Tampa, USA	Phase II	Lenvatinib + pembrolizumab
An Open-Label Phase 2 Study of Surufatinib in Patients With Neuroendocrine Tumours in Europe	Hutchmed Hong Kong, China	Phase II	Surufatinib
Immunotherapy Studies			
Survivin Long Peptide Vaccine in Treating Patients With Metastatic Neuroendocrine Tumors	Rosewell Park Buffalo, USA	Phase I	Survivin peptide vaccine + GM-CSF + sandostatin
PRRT Studies			
Lutathera in People With Gastroenteropancreatic, Bronchial or Unknown Primary Neuroendocrine Tumors That Have Spread to the Liver	Memorial Sloan Kettering Cancer Center New York, USA	Phase I	Intra-arterial 177Lu-DOTATATE
Personalized PRRT of Neuroendocrine Tumors (P-PRRT)	CHU de Quebec-Universite Laval Quebec City, Canada	Phase II	177Lu-DOTATATE dosimetry
A Clinical Study to Assess the Combination of Two Drugs (177Lu-DOTATATE and Nivolumab) in Neuroendocrine Tumors	Fundación de Investigación HM Madrid, Spain	Phase II	177Lu-DOTATATE + nivolumab
Targeted Alpha-emitter Therapy of PRRT Naive Neuroendocrine Tumor Patients (ALPHAMEDIX02)	Radiomedix Houston, USA	Phase II	212Pb-DOTAMTATE
Treatment Using 177Lu-DOTA-EB-TATE in Patients With Advanced Neuroendocrine Tumors	Peking Union Medical College Hospital Beijing, China	Phase I	177Lu-DOTA-EB-TATE
Chemotherapy Studies			
CapTemY90 for Grade 2 NET Liver Metastases (CapTemY90)	Abramson Cancer Center at Penn Medicine Philadelphia, USA	Phase II	Capecitabine/temozolomide and Y90 radioembolization
Miscellaneous Agents			
Study to Evaluate the Safety, PK, and Dose Response of Paltusotine in Subjects With Carcinoid Syndrome	Crinetics Pharmaceuticals San Diego, USA	Phase II	Paltusotine
Study of CVM-1118 for Patients With Advanced Neuroendocrine Tumors	TaiRx, Inc. Taipei City, Taiwan	Phase II	CVM-1118 (TRX-818)

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
