# Peer review of "Emerging Precision Medicine Approaches for Lung Neuroendocrine Tumors"

_cancers, 2023, doi:10.3390/cancers15235575_

Round 1

Reviewer 1 Report

Comments and Suggestions for Authors

The review report on "Emerging Precision Medicine Approaches for Lung Neuroen-2 docrine Tumors" is a well documented draft and with a lot of informations for the readers. The table adds a precise depiction of the entire review which will help the readers. Minor addition/revisions are below:

1. What percentage of population get NETs out of the all the lung cancer patients?

2. What about epigenetic factors responsible for NETs?

3. Is the development and progression of NETs related to age? Where is this most prevalent? It would be good to add a section. 

Author Response

Thank you for these suggestions, which we believe have strengthened the manuscript. Please see below for individual responses.

Reviewer: What percentage of population get NETs out of the all the lung cancer patients?

Response: I have added a sentence on line 43 to indicate that 2% of all lung cancers are lung NETs.

Reviewer: What about epigenetic factors responsible for NETs?

Responses: Epigenetic factors are important for the pathogenesis of NETs. I’ve added to the section describing this, lines 99-103, and also below:

There is also increasing appreciation for the role of epigenetic modifications and dysregulation in driving NET evolution. Hypermethylation of several tumor suppressor genes, such as RASSF1 and p15INK4b, has been described in LNETs, along with loss of histone 4 acetylation at lysine 16 and trimethylation at lysine 20.

Reviewer: Is the development and progression of NETs related to age? Where is this most prevalent? It would be good to add a section.

Response: I have added a section to provide more epidemiological background information about lung NETs (see below), which is now included on lines 43-50 in the manuscript:

LNETs account for approximately 2% of all lung cancers in adults. The lung is the second most common NET primary site after the gastrointestinal tract. LNETs typically develop between the 4th and 6th decade of life, and most series suggest a higher incidence in women compared with men and in White compared with Black or Asian populations. The incidence of LNETs appears to be increasing over time. The reasons for this rising incidence are not entirely clear. Improved cross-sectional imaging and bronchoscopy techniques may be contributing, along with better recognition of the histology by pathologists and increased awareness among clinicians.

Reviewer 2 Report

Comments and Suggestions for Authors

Dear authors, 

Thank you for an interesting paper and a comprehensive review. 

Only few comments

1. The information on SSTR antagonists is a little unclear, could it be more clear what you think of this emerging modality?

2. How are the effects of PDL and PDL1 treatment in LCNEC and SCNEC? Please describe. 

Author Response

Responses to Reviewer 2: Thank you for the instructive comments, please see my responses below.

Reviewer comment 1: The information on SSTR antagonists is a little unclear, could it be more clear what you think of this emerging modality?

Response: Thanks for the feedback. I do think this is a promising emerging modality for both SSTR imaging and radioligand therapy, I’ve added some language to reflect this on lines 346-349.

“By binding more sites on NET cells, SSTR antagonists may further improve treatment responses, particularly when the density of SSTR is low. SSTR antagonists such as 177Lu-satoreotide tetraxetan are now being investigated as a promising new approach, both for imaging and therapeutic purposes in NETs.

Reviewer comment 2: How are the effects of PDL and PDL1 treatment in LCNEC and SCNEC? Please describe.

Response: This is an important question—however, for the purposes of this review article, I chose to focus on well-differentiated lung NETs (typical and atypical carcinoids) and emerging precision medicine approaches for their treatment. I did not include a discussion about treatment of the high-grade neuroendocrine carcinomas of the lung, which have their own unique biology, molecular drivers, risk factors, and separate treatment strategies. In this Cancers special issue, there is another review article written by Shruti Patel and Millie Das that focuses on treatment of small cell lung cancer, which includes a review of the efficacy of PD-1/PD-L1 blockade. I can add a new section if the reviewers deem necessary, but I would advocate for leaving the focus as it currently stands.

Reviewer 3 Report

Comments and Suggestions for Authors

Title: Emerging Precision Medicine Approaches for Lung Neuroen- 2 docrine Tumors

 Author: Claire Mulvey

 This is a well-written review on Lung Neuroendocrine tumors (LNETs) with a special view on precision medicine. The author address with great details the emerging therapeutical approaches used for this specific type of cancer.

Main Points:

Provide more details about origin : “ arise from enterochromaffin cells.” (L33). These cells are present in GI tract, what about lungs? Later (L63) it is mentioned that they can be of pancreatic origin… Needs clarification altogether.

Scheme of progression of LNETs/LNECs/LNENs

Minor Points:

L4. Claire Mulvey 1,2* what does 2 correspond to ?

L14.  “In this review, we review”. Rephrase please

L32 Lung neuroendocrine neoplasms (NENs) rather LNENs (change through whole manuscript)

L62 … Ki-67 proliferation indices. Index ?

L62. “increasing appreciation for the phenomenon (…)”. It sounds strange “challenges of grade progression”?

Table 1. “Natioanl Cancer Institute » typo to correct. Also for all (including NCI) add city and County for sponsor.

L300. « with confirmed survivn expression ». Survivin

Comments on the Quality of English Language

Only few typos were found.

Author Response

Responses to reviewer 3:

Thank you very much for catching these issues. Please see responses to individual suggests below:

Reviewer: L4. Claire Mulvey 1,2* what does 2 correspond to ?

Response: Thank you, this was a typo, I have removed the 2 subscript.

Reviewer: L14.  “In this review, we review”. Rephrase please

Response: I have changed the phrasing to “In this review, we summarize”

Reviewer: L32 Lung neuroendocrine neoplasms (NENs) rather LNENs (change through whole manuscript)

Response: Thank you, as suggested we have changed the abbreviation for lung neuroendocrine neoplasms or tumors to LNEN/LNET throughout the whole manuscript.

Reviewer: L62 … Ki-67 proliferation indices. Index ?

Response: I have changed the wording to Ki-67 proliferation index 

Reviewer: L62. “increasing appreciation for the phenomenon (…)”. It sounds strange “challenges of grade progression”?

Response: Thank you for the suggestion, I’ve changed the wording here to “increasing appreciation for the challenges of grade progression.”

Reviewer: Table 1. “Natioanl Cancer Institute » typo to correct. Also for all (including NCI) add city and County for sponsor.

Response : I have fixed this type and added city and country for all sponsors.

Reviewer: L300. « with confirmed survivn expression ». Survivin

Response: This typo is corrected.